# High Dimensional Sparse Canonical Correlation Analysis for Elliptical Symmetric Distributions

## Abstract

This paper proposes a robust high-dimensional sparse canonical correlation analysis (CCA) method for investigating linear relationships between two high-dimensional random vectors, focusing on elliptical symmetric distributions. Traditional CCA methods, based on sample covariance matrices, struggle in high-dimensional settings, particularly when data exhibit heavy-tailed distributions. To address this, we introduce the spatial-sign covariance matrix as a robust estimator, combined with a sparsity-inducing penalty to efficiently estimate canonical correlations. Theoretical analysis shows that our method is consistent and robust under mild conditions, converging at an optimal rate even in the presence of heavy tails. Simulation studies demonstrate that our approach outperforms existing sparse CCA methods, particularly under heavy-tailed distributions. A real-world application further confirms the method's robustness and efficiency in practice. Our work provides a novel solution for high-dimensional CCA, offering significant advantages over traditional methods in terms of both stability and performance.

*Keywords*: Canonical correlation analysis, Elliptical symmetric distributions, High dimensional data, Spatial-sign

## 1 Introduction

Canonical correlation analysis (CCA) is a fundamental multivariate statistical technique that explores the linear relationships between two sets of variables. It has been widely applied across diverse fields such as biomedical research, neuroimaging, and genomics (Hardoon et al., 2004; Chi et al., 2013; Safo et al., 2018). By identifying maximally correlated linear combinations between paired datasets, CCA serves as a powerful tool for uncovering complex cross-domain associations and facilitating integrative data analysis.

Despite its wide applicability, classical sample covariance-based CCA often faces substantial limitations in modern data settings. Two primary challenges hinder its effectiveness: the high dimensionality of contemporary datasets and deviations from the multivariate normality assumption. In high-dimensional scenarios—where the number of variables exceeds, or is comparable to, the sample size—the sample covariance matrices involved in CCA become ill-conditioned or singular, rendering traditional CCA unstable or even inapplicable (Hardoon et al., 2004; Guo et al., 2016). Moreover, when the underlying data distributions deviate from normality, as is common in genomics or financial data, the performance of standard CCA deteriorates due to its sensitivity to outliers and heavy tails.

To address these issues, a growing body of research has proposed robust and regularized extensions of CCA that incorporate sparsity assumptions, shrinkage techniques, such as González et al. (2008); Parkhomenko et al. (2009); Witten et al. (2009); Chen & Liu (2011); Chi et al. (2013); Cruz Cano & Lee (2014); Gao et al. (2015); Wilms & Croux (2015); Gao et al. (2017); Safo et al. (2018). These modern adaptations aim to improve estimation accuracy, enhance interpretability, and ensure reliable inference in high-dimensional settings. As one of the most popular sparse CCA methods, Witten et al. (2009) proposed the penalized matrix decomposition (PMD) that replaces the sample

covariance matrix of two random vectors with identity matrices to avoid singularity. However, Chen et al. (2017) showed that the sparse CCA directions from PMD may be inconsistent when the covariance matrix of these two vectors are far from diagonal. So they relaxed the diagonal assumption by assuming the sparsity of the covariance matrix. Gao et al. (2017) proposed a convex programming with group lasso refinement, which does not impose any assumption on the covariance matrix or precision matrix and achieves the minimax estimation risk (Gao et al., 2015). Mai & Zhang (2019) proposed an iterative penalized least square approach to sparse CCA, which does not need the sparse assumption of the covariance matrix, either.

To address the limitations of classical CCA under nonnormal data, various improved methods have been developed. Nonparametric approaches, such as kernel CCA (Hardoon et al., 2004), leverage kernel techniques to capture nonlinear associations without distributional assumptions. On the other hand, parametric and semiparametric methods offer probabilistic frameworks for CCA. For example, Zoh et al. (2016) proposed a probabilistic CCA tailored for count data. Agniel & Cai (2017) introduced a semiparametric normal transformation for analyzing mixed-type variables, which involves nonparametric maximum likelihood estimation of marginal transformation functions. In addition, there exists a substantial body of research on robust covariance matrix estimators designed to be resistant to outliers and heavy-tailed distributions, which can enhance the reliability of classical CCA based on Pearson correlations. Notable examples include the minimum covariance determinant (MCD) (Rousseeuw, 1984), the S-estimator (Lopuhaa, 1989), and Tyler's M-estimator (Tyler, 1987). Several studies have explored incorporating these robust covariance estimators into CCA frameworks or replacing Pearson correlation with more robust association measures. Many of these approaches rely on eigen-decomposition of robust covariance or correlation matrices (Alfons et al., 2017a; Branco et al., 2005; Taskinen et al., 2006; Visuri et al., 2003); however, their applicability may be limited when the data lack finite moments, posing challenges for consistency and interpretation. Yoon et al. (2020) derived rank-based estimator instead of the sample correlation matrix within the sparse CCA framework motivated by Chi et al. (2013) and Wilms & Croux (2015).

To overcome these challenges of high dimension and non-normality simultaneously, we propose a robust high-dimensional sparse canonical correlation analysis method, based on the spatial-sign covariance matrix (Oja, 2010). The spatial-sign covariance matrix, which is a robust estimator for covariance, is well-suited for elliptical symmetric distributions and offers superior performance in high-dimensional settings with heavy-tailed distributions. Raninen et al. (2021), Raninen & Ollila (2021), and Ollila & Breloy (2022) proposed a series of linear shrinkage estimators based on spatial-sign covariance matrices and showed their advantages over the existing methods based on sample covariance matrix. Feng (2024) considered high dimensional sparse principal component analysis via spatial-sign covariance matrix. Lu & Feng (2025) also proposed a high dimensional precision matrix estimator based on spatial-sign covariance matrix and applied it to elliptical graphic model and linear discriminant analysis.

In this article, we propose a spatial-sign based sparse canonical correlation analysis via $l_1$ penalty. We establish the theoretical properties of the proposed method, demonstrating its consistency and robustness under mild regularity conditions. Specifically, we show that the new estimator converges at an optimal rate and remains robust to deviations from normality, such as the presence of heavy tails in the data. The proposed method is compared with existing sparse CCA techniques, which typically rely on Gaussian or sub-Gaussian assumptions. Simulation studies confirm that our method outperforms these alternatives, especially under heavy-tailed conditions, in terms of both accuracy and stability. Moreover, we apply our method to a real-world data set. The results further illustrate the robustness and efficiency of our approach in practical scenarios.

The remainder of the article is organized as follows. Section 2 introduces the spatial-sign-based sparse CCA method and presents its theoretical properties. Section 3 provides simulation studies to evaluate the performance of the proposed approach, and Section 4 illustrates its practical utility through a real data application. All technical details and proofs are provided in the Appendix.

## 2 SPATIAL-SIGN BASED SPARSE CANONICAL CORRELATION ANALYSIS

Let $\boldsymbol{X}_1 \in \mathbb{R}^{p_1}$ and $\boldsymbol{X}_2 \in \mathbb{R}^{p_2}$ be two random vectors with covariance matrices $\boldsymbol{\Sigma}_1 = \text{cov}(\boldsymbol{X}_1)$, $\boldsymbol{\Sigma}_2 = \text{cov}(\boldsymbol{X}_2)$, and cross-covariance matrix $\boldsymbol{\Sigma}_{12} = \text{cov}(\boldsymbol{X}_1, \boldsymbol{X}_2)$. Classical CCA aims to identify linear projections $\boldsymbol{w}_1^\top \boldsymbol{X}_1$ and $\boldsymbol{w}_2^\top \boldsymbol{X}_2$ that achieve the highest possible correlation (Hotelling, 1936).

Formally, it solves the optimization problem:

$$\underset{\boldsymbol{w}_1, \boldsymbol{w}_2}{\text{maximize}} \left\{ \boldsymbol{w}_1^\top \boldsymbol{\Sigma}_{12} \boldsymbol{w}_2 \right\} \quad \text{s.t.} \quad \boldsymbol{w}_1^\top \boldsymbol{\Sigma}_1 \boldsymbol{w}_1 = 1, \quad \boldsymbol{w}_2^\top \boldsymbol{\Sigma}_2 \boldsymbol{w}_2 = 1, \tag{1}$$

with a closed-form solution via the singular value decomposition of the normalized cross-covariance matrix $\boldsymbol{\Sigma}_1^{-1/2} \boldsymbol{\Sigma}_{12} \boldsymbol{\Sigma}_2^{-1/2}$.

In practice, these population quantities are estimated by their sample counterparts. However, in high-dimensional scenarios where the number of variables exceeds the sample size, sample covariance matrices $\hat{\boldsymbol{\Sigma}}_1$ and $\hat{\boldsymbol{\Sigma}}_2$ become singular, rendering the classical solution unstable or undefined. To address this, sparse CCA formulations introduce $\ell_1$ penalties on $\boldsymbol{w}_1$ and $\boldsymbol{w}_2$ to induce sparsity and avoid overfitting (Parkhomenko et al., 2009; Witten et al., 2009; Chi et al., 2013; Wilms & Croux, 2015):

$$\underset{\boldsymbol{w}_1, \boldsymbol{w}_2}{\text{maximize}} \left\{ \boldsymbol{w}_1^\top \hat{\boldsymbol{\Sigma}}_{12} \boldsymbol{w}_2 - \lambda_1 \|\boldsymbol{w}_1\|_1 - \lambda_2 \|\boldsymbol{w}_2\|_1 \right\} \quad \text{s.t.} \quad \boldsymbol{w}_1^\top \hat{\boldsymbol{\Sigma}}_1 \boldsymbol{w}_1 \le 1, \quad \boldsymbol{w}_2^\top \hat{\boldsymbol{\Sigma}}_2 \boldsymbol{w}_2 \le 1.$$

The use of inequality constraints ensures convexity, which is advantageous for optimization.

Despite its success in high-dimensional contexts, this framework still relies on second-order moment assumptions and can perform poorly when data exhibit heavy tails or outliers. This limitation has motivated recent developments that replace the sample covariance matrices with robust alternatives—such as rank-based (Yoon et al., 2020) or spatial-sign-based estimators—allowing for improved performance in non-Gaussian environments.

Suppose $\boldsymbol{X} = (\boldsymbol{X}_1^\top, \boldsymbol{X}_2^\top)^\top$ are generated from the elliptical symmetric distribution $\boldsymbol{X} \sim E_p(\boldsymbol{\mu}, \boldsymbol{\Sigma}, r)$, i.e.

$$\boldsymbol{X} = \boldsymbol{\mu} + r\boldsymbol{\Gamma}\boldsymbol{u}, \tag{2}$$

where $\boldsymbol{u}$ is uniformly distributed on the sphere $\mathbb{S}^{p-1}$ and $r$ is a scalar random variable with $E(r^2) = p$ and independent with $\boldsymbol{u}$. So the covariance matrix of $\boldsymbol{X}$ is $\boldsymbol{\Sigma} = \boldsymbol{\Gamma}\boldsymbol{\Gamma}^\top$. The spatial-sign covariance matrix $\mathbf{S}$ is defined as: $\mathbf{S} = E\left(U(\boldsymbol{X} - \boldsymbol{\mu})U(\boldsymbol{X} - \boldsymbol{\mu})^\top\right)$, where $U(\boldsymbol{x}) = \frac{\boldsymbol{x}}{\|\boldsymbol{x}\|}I(\boldsymbol{x} \neq 0)$. Suppose we observe a set of independent and identically distributed samples $\{\boldsymbol{X}_1, \ldots, \boldsymbol{X}_n\}$ drawn from the model specified in equation (2). Then, the corresponding sample spatial-sign covariance matrix $\hat{\mathbf{S}}$ is defined as $\hat{\mathbf{S}} = \frac{1}{n}\sum_{i=1}^n U(\boldsymbol{X}_i - \hat{\boldsymbol{\mu}})U(\boldsymbol{X}_i - \hat{\boldsymbol{\mu}})^\top$ where $\hat{\boldsymbol{\mu}}$ is the sample spatial-median, i.e. $\hat{\boldsymbol{\mu}} = \arg\min_{\boldsymbol{\mu} \in \mathbb{R}^p} \sum_{i=1}^n \|\boldsymbol{X}_i - \boldsymbol{\mu}\|_2$. Let $\boldsymbol{X}_i = (\boldsymbol{X}_{i1}^\top, \boldsymbol{X}_{i2}^\top)^\top$ and define

$$\hat{\mathbf{S}} = \begin{pmatrix} \hat{\mathbf{S}}_1 & \hat{\mathbf{S}}_{12} \\ \hat{\mathbf{S}}_{21} & \hat{\mathbf{S}}_2 \end{pmatrix},$$

where $\hat{\mathbf{S}}_1 \in \mathbb{R}^{p_1 \times p_1}, \hat{\mathbf{S}}_2 \in \mathbb{R}^{p_2 \times p_2}$ and $\hat{\mathbf{S}}_{12} \in \mathbb{R}^{p_1 \times p_2}$.

As shown in Lemma 6 in Lu & Feng (2025), the covariance matrix $\boldsymbol{\Sigma}$ could be well approximated by $\text{tr}(\boldsymbol{\Sigma})\mathbf{S}$ as the dimension goes to infinity. So we replace the sample covariance matrix with the sample spatial-sign covariance matrix and reformulate the problem as

$$\underset{\boldsymbol{w}_1, \boldsymbol{w}_2}{\text{maximize}} \left\{ \boldsymbol{w}_1^\top p\hat{\mathbf{S}}_{12}\boldsymbol{w}_2 - \lambda_1 \|\boldsymbol{w}_1\|_1 - \lambda_2 \|\boldsymbol{w}_2\|_1 \right\} \quad \text{s.t.} \ \boldsymbol{w}_1^\top p\hat{\mathbf{S}}_1 \boldsymbol{w}_1 \leqslant 1, \boldsymbol{w}_2^\top p\hat{\mathbf{S}}_2 \boldsymbol{w}_2 \leqslant 1. \tag{3}$$

Here, it is unnecessary to estimate $\text{tr}(\boldsymbol{\Sigma})$ because scaling the linear combinations by positive constants does not affect the canonical correlation. Specifically, for any $c_1 > 0$ and $c_2 > 0$, we have $\text{cor}(c_1\boldsymbol{w}_1^\top \boldsymbol{X}_1, c_2\boldsymbol{w}_2^\top \boldsymbol{X}_2) = \text{cor}(\boldsymbol{w}_1^\top \boldsymbol{X}_1, \boldsymbol{w}_2^\top \boldsymbol{X}_2)$. So, without loss of generality, we directly assume that $\text{tr}(\boldsymbol{\Sigma}_1) = p_1$ and $\text{tr}(\boldsymbol{\Sigma}_2) = p_2$. We refer to the estimation procedure obtained by solving (3) as Spatial-Sign based Sparse Canonical Correlation Analysis, abbreviated as SSCCA.

The tuning parameters $\lambda_1, \lambda_2$ are selected similar to the BIC method proposed in Yoon et al. (2020). Define

$$f(\tilde{\boldsymbol{w}}_1) = \tilde{\boldsymbol{w}}_1^\top p\hat{\mathbf{S}}_1 \tilde{\boldsymbol{w}}_1 - 2\tilde{\boldsymbol{w}}_1^\top p\hat{\mathbf{S}}_{12}\boldsymbol{w}_2 + \boldsymbol{w}_2 p\hat{\mathbf{S}}_2 \boldsymbol{w}_2$$

for the residual sum of squares. Furthermore, motivated by the performance of the adjusted degrees of freedom variance estimator in Reid et al. (2016), we also propose the following two criteria

$$\text{BIC}_1 = f(\tilde{\boldsymbol{w}}_1) + \text{df}_{\tilde{\boldsymbol{w}}_1} \frac{\log n}{n}, \quad \text{BIC}_2 = \log\left\{ \frac{n}{n - \text{df}_{\tilde{\boldsymbol{w}}_1}} f(\tilde{\boldsymbol{w}}_1) \right\} + \text{df}_{\tilde{\boldsymbol{w}}_1} \frac{\log n}{n}.$$

Here $\mathrm{df}_{\tilde{w}_1}$ coincides with the size of the support of $\tilde{w}_1$ (Tibshirani & Taylor, 2012). The criteria for $w_2$ are defined analogously to those for $w_1$.

We use the same algorithm proposed by Yoon et al. (2020) to solve our problem (3). Here we directly use the function *find_w12bic* in R package `mixedCCA` to solve the problem (3) and the tuning parameters are generated by the function *lambdaseq_generate* in R package `mixedCCA`. The sample spatial-sign covariance matrix is estimated using the function *SCov* in R package `SpatialNP`.

We now proceed to establish the theoretical properties of the proposed estimators. To start with, we have the following lemma on the estimation accuracy of the covariance matrix estimate $p\hat{\mathbf{S}}$.

**Lemma 1.** *Assume that (i) The eigenvalues of $\boldsymbol{\Sigma}$ are bounded such that $\kappa^{-1/2} \leq \lambda_{\min}(\boldsymbol{\Sigma}) \leq \lambda_{\max}(\boldsymbol{\Sigma}) \leq \kappa^{1/2}$; (ii) The random variable $r$ in equation (2) satisfies that $\mathbb{E}(|r|^{-1}) \leq \zeta$ and $\mathbb{E}(|r|^{-k})/\{\mathbb{E}(|r|^{-1})\}^k \leq \zeta$ for $k = 2, 3, 4$ and constant $\zeta > 0$ and $r^{-1}$ is sub-Gaussian distributed. Then with probability at least $1 - \alpha$,*

$$\|p\hat{\mathbf{S}} - \boldsymbol{\Sigma}\|_\infty \leq C\sqrt{\frac{\log p + \log(\alpha^{-1/2})}{n}} + \frac{C}{\sqrt{p}}.$$

Assumption (i) in Lemma 1 implies that the covariance $\boldsymbol{\Sigma}$ is both lower and upper bounded. Assumption (ii) in Lemma 1 ensures that the norm $\|\boldsymbol{X} - \boldsymbol{\mu}\|$ is bounded away from zero, which makes the spatial-sign function $U(\cdot)$ well-behaved. This assumption is mild because for high-dimensional data, it is widely observed that the norm of the sample vectors $\|\boldsymbol{X} - \boldsymbol{\mu}\| \approx \Theta(\sqrt{p})$. Similar assumptions and the proof of Lemma 1 can be found in Lu & Feng (2025).

Let $\boldsymbol{w}_{1,*}, \boldsymbol{w}_{2,*}$ be the maximizer of the maximization problem (1) and $\rho_{1,*} = \boldsymbol{w}_{1,*}^\top \boldsymbol{\Sigma}_{12} \boldsymbol{w}_{2,*}$. We also need the following assumptions for the sparse CCA task.

**Assumption 1** (Bounded condition number). *The covariance matrices enjoy a bounded restricted condition number such that*

$$\frac{\sup_{\|\boldsymbol{w}\|_1 \leq C\tau_1, \|\boldsymbol{w}\|_2 = 1} \boldsymbol{w}^\top \Sigma_j \boldsymbol{w}}{\inf_{\|\boldsymbol{w}\|_1 \leq C\tau_1, \|\boldsymbol{w}\|_2 = 1} \boldsymbol{w}^\top \Sigma_j \boldsymbol{w}} \leq \kappa, \forall j = 1, 2,$$

*for some constants $C, \tau_1, \kappa > 0$.*

**Assumption 2** (Gap of leading canonical correlation). *There exists $c > 0$ such that $\rho_1^* - \max_{k \geq 2} \rho_k^* > \gamma$.*

**Assumption 3** (Sparsity of canonical correlation vector). *$\max\{\|\boldsymbol{w}_{1,*}\|_1, \|\boldsymbol{w}_{2,*}\|_1\} \leq \tau_1$.*

Assumption 1 is widely used in high dimensional data analysis (Mai & Zhang, 2019), which also holds under Assumption (i) in Lemma 1. Assumption 2 ensures the identifiability of the leading canonical correlation component. Assumption 3 restricts the true left and right canonical correlation vectors to the $L_1$ core. If we further assume that both $\boldsymbol{w}_{1,*}$ and $\boldsymbol{w}_{2,*}$ are $s$-sparse and all the entries of them are bounded, then $\tau_1 = O(s)$. Similar assumptions on the sparsity can be found in the literature on sparse PCA (Zou & Xue, 2018) and sparse CCA (Mai & Zhang, 2019).

Denote $\epsilon = C[\sqrt{\{\log p + \log(\alpha^{-1/2})\}/n} + p^{-1/2}]$ as the estimation error bound in Lemma 1. We have the following results on the performance of the robust SSCCA estimate.

**Theorem 1.** *Assume the error bound in Lemma 1 and Assumption 3 hold and choose $\lambda = C_1 \epsilon \tau_1$ for some constant $C_1 > 0$. With probability at least $1 - \alpha$, there exists a local maximizer $(\hat{\boldsymbol{w}}_1, \hat{\boldsymbol{w}}_2)$ of the nonconvex problem (3) which satisfies that,*

$$\hat{\boldsymbol{w}}_1^\top \boldsymbol{\Sigma}_{12} \hat{\boldsymbol{w}}_2 \geq (1 + \tau_1^2 \epsilon)^{-1} \rho_{1,*} - \{(1 - \epsilon\tau_1^2)^{-1} + 2C_1(1 - \epsilon\tau_1^2)^{-1/2} + C_2\}\tau_1^2 \epsilon; \quad (4)$$

$$\frac{\hat{\boldsymbol{w}}_1^\top \boldsymbol{\Sigma}_{12} \hat{\boldsymbol{w}}_2}{\sqrt{\hat{\boldsymbol{w}}_1^\top \boldsymbol{\Sigma}_1 \hat{\boldsymbol{w}}_1} \sqrt{\hat{\boldsymbol{w}}_2^\top \boldsymbol{\Sigma}_2 \hat{\boldsymbol{w}}_2}} \geq \frac{(1 + \tau_1^2 \epsilon)^{-1} \rho_{1,*} - \{(1 - \tau_1^2 \epsilon)^{-1} + 2C_1(1 - \tau_1^2 \epsilon)^{-1/2} + C_2\}\tau_1^2 \epsilon}{1 + C_2 \tau_1^2 \epsilon},$$

$$(5)$$

*where the constants $C_1$ and $C_2$ only depend on the parameter $\tau_1$, $\rho_{1,*}$ and $\epsilon$. Additionally assume Assumptions 1 and 2 hold, we have,*

$$\min_{j \in \{1,2\}} \cos^2(\angle(\hat{\boldsymbol{w}}_j, \boldsymbol{w}_{j,*}))$$

$$\geq 1 - \frac{4\kappa}{\gamma}\left\{\frac{(1 + C_2)\tau_1^2 \epsilon + C_2 \tau_1^4 \epsilon^2}{(1 + \tau_1^2 \epsilon)(1 + C_2 \tau_1^2 \epsilon)}\rho_{1,*} + \frac{\{(1 - \tau_1^2 \epsilon)^{-1} + 2C_1(1 - \tau_1^2 \epsilon)^{-1/2} + C_2\}\tau_1^2 \epsilon}{1 + C_2 \tau_1^2 \epsilon}\right\}.$$

Consequently, if we assume $\tau_1^2 \left( \sqrt{(\log p)/n} + p^{-1/2} \right) \to 0$, then it follows that with high probability, $\cos^2 \left( \angle(\hat{\boldsymbol{w}}_j, \boldsymbol{w}_{j,*}) \right) \to 1$ for $j = 1, 2$, which establishes the consistency of our proposed method. It is worth noting that the convergence rate derived here differs slightly from those in Yoon et al. (2020) and Mai & Zhang (2019), where the term $p^{-1/2}$ does not appear. This discrepancy arises from the approximation bias introduced when using $p\mathbf{S}$ to estimate $\boldsymbol{\Sigma}$. However, this bias term becomes negligible when $p \log p/n \to \infty$, effectively aligning the convergence behavior with that in the aforementioned works.

## 3  SIMULATION

In this section, we compare the performance of our proposed method, SSCCA, with two existing approaches: the method by Yoon et al. (2020), referred to as KSCCA, and the method by Chi et al. (2013), referred to as SCCA. For a fair comparison, all three methods are implemented using the same convex optimization algorithm introduced in Yoon et al. (2020). The key distinction among the methods lies in the choice of covariance matrix estimators employed in the analysis.

We consider the case when $\boldsymbol{\Sigma}_1$ and $\boldsymbol{\Sigma}_2$ are block diagonal matrices with five blocks, each of dimension $d/5 \times d/5$, where the $(i, j)$-th element of each block takes value $0.8^{|i-j|}$. Here $d = p_1 = p_2 = p/2$. Similar to Tan et al. (2018), we consider two cases of $\boldsymbol{\Sigma}_{12}$:

(I) Low Rank

$$\boldsymbol{\Sigma}_{12} = \boldsymbol{\Sigma}_1 \boldsymbol{w}_1^* \varphi_1 \left( \boldsymbol{w}_2^* \right)^\top \boldsymbol{\Sigma}_2$$

where $\varphi_1 = 0.9$ is the largest generalized eigenvalue and $\boldsymbol{w}_1^*$ and $\boldsymbol{w}_2^*$ are the leading pair of canonical directions. Here $\boldsymbol{w}_j^* = \boldsymbol{v}/\sqrt{\boldsymbol{v}^T \boldsymbol{\Sigma}_j \boldsymbol{v}}, j = 1, 2$ where $\boldsymbol{v} = (v_1, \cdots, v_d)^T$ and $v_k = 1/\sqrt{3}$ for $k = 1, 6, 11$ and $v_k = 0$ otherwise.

(II) Approximately Low Rank

$$\boldsymbol{\Sigma}_{12} = \boldsymbol{\Sigma}_1 \boldsymbol{w}_1^* \varphi_1 \left( \boldsymbol{w}_2^* \right)^\top \boldsymbol{\Sigma}_2 + \boldsymbol{\Sigma}_1 \boldsymbol{W}_1^* \boldsymbol{\Lambda} \left( \boldsymbol{W}_2^* \right)^\top \boldsymbol{\Sigma}_2$$

where $\varphi_1, \boldsymbol{w}_1^*, \boldsymbol{w}_2^*$ are the same as (I). Additionally, $\boldsymbol{\Lambda} \in \mathbb{R}^{50 \times 50}$ is a diagonal matrix with diagonal entries $0.1$, and $\boldsymbol{W}_1^*, \boldsymbol{W}_2^* \in \mathbb{R}^{d \times 50}$ are normalized orthogonal matrices such that $\left( \boldsymbol{W}_1^* \right)^\top \boldsymbol{\Sigma}_1 \boldsymbol{W}_1^* = \mathbf{I}$ and $\left( \boldsymbol{W}_2^* \right)^\top \boldsymbol{\Sigma}_2 \boldsymbol{W}_2^* = \mathbf{I}$.

The data consists of two $n \times d$ matrices $\mathbf{X}$ and $\mathbf{Y}$. We consider three elliptical distributions:

(i) Multivarite Normal Distribution: $X_i \sim N(\boldsymbol{\mu}, \boldsymbol{\Sigma})$;

(ii) Multivariate $t$-distribution: $X_i \sim t(\boldsymbol{\mu}, \boldsymbol{\Sigma}, 3)/\sqrt{3}$;

(iii) Mixture of multivariate Normal distribution: $X_i \sim MN(\boldsymbol{\mu}, \boldsymbol{\Sigma}, 10, 0.8)/\sqrt{20.8}$;

Here $t_p(0, \boldsymbol{\Sigma}, v)$ denotes a $p$-dimensional $t$-distribution with degrees of freedom $v$ and scatter matrix $\boldsymbol{\Sigma}$. $MN(\boldsymbol{\mu}, \boldsymbol{\Sigma}, \kappa, \gamma)$ refers to a mixture multivariate normal distribution with density function $(1 - \gamma) f_p(\mathbf{0}, \boldsymbol{\Sigma}) + \gamma f_p \left( \mathbf{0}, \kappa^2 \boldsymbol{\Sigma} \right)$, where $f_p(\boldsymbol{a}, \mathbf{B})$ is the density function of the $p$-dimensional normal distribution with mean $\boldsymbol{a}$ and covariance matrix $\mathbf{B}$. We consider three sample sizes $n = 100, 200, 300$ and two different dimensions $p = 400, 800$. All the results are based on 1000 replications and are executed on an Ubuntu 20.04 LTS server with 64 Intel(R) Xeon(R) CPU E5-2690 v4 @ 2.60GHz (56 cores), 128G RAM and the R platform with version 4.3.1.

To compare the performance of the methods, we evaluate the expected out-of-sample correlation

$$\hat{\rho} = \left| \frac{\hat{\boldsymbol{w}}_1^\top \boldsymbol{\Sigma}_{12} \hat{\boldsymbol{w}}_2}{\left( \hat{\boldsymbol{w}}_1^\top \boldsymbol{\Sigma}_1 \hat{\boldsymbol{w}}_1 \right)^{1/2} \left( \hat{\boldsymbol{w}}_2^\top \boldsymbol{\Sigma}_2 \hat{\boldsymbol{w}}_2 \right)^{1/2}} \right|,$$

and the prediction loss

$$L \left( \boldsymbol{w}_g^*, \hat{\boldsymbol{w}}_g \right) = 1 - \frac{\left| \hat{\boldsymbol{w}}_g^\top \boldsymbol{\Sigma}_g \boldsymbol{w}_g^* \right|}{\left( \hat{\boldsymbol{w}}_g^\top \boldsymbol{\Sigma}_g \hat{\boldsymbol{w}}_g \right)^{1/2}} \quad (g = 1, 2),$$

a similar loss is used in Gao et al. (2017). By the definition of the true canonical correlation $\rho$, for any $\hat{\boldsymbol{w}}_1$ and $\hat{\boldsymbol{w}}_2$ we have that $\hat{\rho} \leqslant \rho$, with equality when $\hat{\boldsymbol{w}}_1 = \boldsymbol{w}_1^*$ and $\hat{\boldsymbol{w}}_2 = \boldsymbol{w}_2^*$. Since $\boldsymbol{w}_g^{*\top} \boldsymbol{\Sigma}_g \boldsymbol{w}_g^* = 1, L\left(\boldsymbol{w}_g^*, \hat{\boldsymbol{w}}_g\right) \in [0, 1]$ with $L\left(\boldsymbol{w}_g^*, \hat{\boldsymbol{w}}_g\right) = 0$ if $\hat{\boldsymbol{w}}_g = \boldsymbol{w}_g^*$. We also evaluate the variable selection performance using false positive and false negative rates, defined respectively as

$$\text{FPR}_g = 1 - \frac{\#\left\{j : \hat{\boldsymbol{w}}_{gj} \neq 0, \boldsymbol{w}_{gj}^* \neq 0\right\}}{\#\left\{j : \boldsymbol{w}_{gj}^* \neq 0\right\}}, \quad \text{FNR}_g = 1 - \frac{\#\left\{j : \hat{\boldsymbol{w}}_{gj} = 0, \boldsymbol{w}_{gj}^* = 0\right\}}{\#\left\{j : \boldsymbol{w}_{gj}^* = 0\right\}} \quad (g = 1, 2).$$

Tables 1–3 present the average estimation error $|\hat{\rho} - \rho|$, prediction loss, and false positive and negative rates under Model I. Similarly, for Model II, the corresponding results are summarized in Tables 4–6. Under the multivariate normal distribution setting, all three methods—SSCCA, KSCCA and SCCA exhibit comparable performance across all evaluation metrics, including estimation accuracy, prediction loss, and variable selection consistency. This is expected since classical covariance-based estimators perform well under Gaussian assumptions, where outliers and extreme values are rare.

However, when the data deviates from normality and follows heavy-tailed distributions, substantial differences emerge. Specifically, SSCCA demonstrates superior performance in both estimation and prediction tasks. This improvement is attributed to the robustness of the spatial-sign covariance matrix, which is less sensitive to outliers and heavy-tailed noise. By leveraging the distributional properties of elliptical distributions, SSCCA effectively captures the underlying correlation structure even when the data contains large deviations or lacks finite higher-order moments. In comparison, KSCCA, which utilizes Kendall's tau-based covariance estimation, performs better than SCCA under heavy-tailed scenarios but is still less robust than SSCCA. While Kendall's tau is more resistant to non-normality than Pearson correlation, it can still be influenced by extreme values, particularly when the tail heaviness is significant. On the other hand, SCCA, which relies on the conventional covariance matrix, suffers greatly in these settings. Its performance deteriorates due to the unreliability and instability of the sample covariance matrix in the presence of heavy-tailed observations.

Overall, the empirical results across multiple simulation settings consistently highlight the advantages of SSCCA in non-Gaussian environments. Its ability to maintain high estimation accuracy, low prediction loss, and accurate variable selection under a variety of distributional settings makes it a powerful and reliable tool for high-dimensional data analysis, particularly when robustness to heavy tails and outliers is critical.

Table 1: The average of absolute difference between expected out-of-sample correlation and true canonical correlation ($|\hat{\rho} - \rho|$) of each method (multiplied by 100) under Model I.

| $n$ | $p$ | Normal Distribution | | | $t_3$ Distribution | | | Mixture Normal Distribution | | |
|---|---|---|---|---|---|---|---|---|---|---|
| | | SCCA | KSCCA | SSCCA | SCCA | KSCCA | SSCCA | SCCA | KSCCA | SSCCA |
| 100 | 400 | 2.7 | 5.7 | 2.8 | 44.7 | 23.4 | 6.7 | 47.1 | 38.1 | 8.7 |
| 200 | 400 | 0.7 | 1.2 | 0.7 | 18.3 | 2.4 | 0.8 | 9.7 | 4.3 | 0.7 |
| 300 | 400 | 0.4 | 0.8 | 0.4 | 9.9 | 1.3 | 0.4 | 4.5 | 2.4 | 0.4 |
| 100 | 800 | 12.6 | 18.7 | 12.8 | 54.6 | 40.3 | 23.7 | 64.5 | 69.1 | 34.5 |
| 200 | 800 | 0.7 | 1.3 | 0.7 | 18.2 | 2.5 | 0.7 | 19.9 | 9.9 | 3.4 |
| 300 | 800 | 0.4 | 0.8 | 0.4 | 14.8 | 1.3 | 0.4 | 5.2 | 2.7 | 0.4 |

## 4 REAL DATA APPLICATION

In this section, we demonstrate the application of our proposed SSCCA method to the "nutrimouse" data set which was generously provided by Pascal Martin from the Toxicology and Pharmacology Laboratory. This data set originates from a nutrigenomic study in mice (Martin et al., 2007) and is publicly available in the R package `CCA` (González et al., 2008). In this data set, two distinct but biologically related sets of variables are available for 40 mice. The first set of variables contains the expression measurements of 120 hepatic genes potentially implicated in nutritional pathways. The second set of variables is comprised of concentrations of 21 hepatic fatty acids. Furthermore, the mice are stratified by two factors: genotypes (wild-type and PPAR$\alpha$ deficient mice) and diets (reference diet "REF", hydrogenated coconut oil diet "COC", sunflower oil diet "SUN", linseed

Table 2: The average prediction loss of each method (multiplied by 100) under Model I.

| | | SCCA | | KSCCA | | SSCCA | |
|---|---|---|---|---|---|---|---|
| $n$ | $p$ | $L(\boldsymbol{w}_1,\hat{\boldsymbol{w}}_1)$ | $L(\boldsymbol{w}_2,\hat{\boldsymbol{w}}_2)$ | $L(\boldsymbol{w}_1,\hat{\boldsymbol{w}}_1)$ | $L(\boldsymbol{w}_2,\hat{\boldsymbol{w}}_2)$ | $L(\boldsymbol{w}_1,\hat{\boldsymbol{w}}_1)$ | $L(\boldsymbol{w}_2,\hat{\boldsymbol{w}}_2)$ |
| | | Normal Distribution | | | | | |
| 100 | 400 | 1.5 | 1.6 | 2.3 | 2.3 | 1.5 | 1.7 |
| 200 | 400 | 0.4 | 0.4 | 0.7 | 0.7 | 0.4 | 0.4 |
| 300 | 400 | 0.2 | 0.2 | 0.4 | 0.4 | 0.2 | 0.3 |
| 100 | 800 | 12.5 | 12.5 | 13.6 | 13.3 | 12.8 | 12.6 |
| 200 | 800 | 0.4 | 0.4 | 0.7 | 0.7 | 0.4 | 0.4 |
| 300 | 800 | 0.2 | 0.2 | 0.5 | 0.4 | 0.2 | 0.2 |
| | | $t_3$ Distribution | | | | | |
| 100 | 400 | 37 | 38 | 8.3 | 8.1 | 4.7 | 4.4 |
| 200 | 400 | 15.9 | 16.1 | 1.3 | 1.4 | 0.4 | 0.5 |
| 300 | 400 | 8.1 | 8.3 | 0.7 | 0.7 | 0.2 | 0.2 |
| 100 | 800 | 50.7 | 51.8 | 30.4 | 30 | 25.1 | 24.7 |
| 200 | 800 | 15.1 | 16.2 | 1.3 | 1.5 | 0.3 | 0.4 |
| 300 | 800 | 12.4 | 12.7 | 0.7 | 0.7 | 0.2 | 0.2 |
| | | Mixture Normal Distribution | | | | | |
| 100 | 400 | 30.6 | 26 | 15.2 | 15.4 | 8.4 | 8.3 |
| 200 | 400 | 6 | 5.3 | 2.3 | 2.5 | 0.4 | 0.4 |
| 300 | 400 | 2.4 | 2.7 | 1.4 | 1.3 | 0.2 | 0.2 |
| 100 | 800 | 57.8 | 55.5 | 47.8 | 48 | 36.8 | 37.1 |
| 200 | 800 | 11.8 | 13.4 | 6.6 | 6.9 | 3.3 | 3.4 |
| 300 | 800 | 3.1 | 2.9 | 1.5 | 1.5 | 0.2 | 0.2 |

Table 3: The average false positive and false negative rates (multiplied by 100) of the selected model size of each method under Model I.

| | | SCCA | | | | KSCCA | | | | SSCCA | | | |
|---|---|---|---|---|---|---|---|---|---|---|---|---|---|
| $n$ | $p$ | $FPR_1$ | $FNR_1$ | $FPR_2$ | $FNR_2$ | $FPR_1$ | $FNR_1$ | $FPR_2$ | $FNR_2$ | $FPR_1$ | $FNR_1$ | $FPR_2$ | $FNR_2$ |
| | | Normal Distribution | | | | | | | | | | | |
| 100 | 400 | 1.3 | 0.8 | 1 | 0.9 | 1.7 | 1.2 | 1 | 1.2 | 1 | 0.8 | 1.7 | 0.9 |
| 200 | 400 | 0 | 0.8 | 0 | 0.8 | 0 | 1.3 | 0 | 1.2 | 0 | 0.7 | 0 | 0.8 |
| 300 | 400 | 0 | 0.7 | 0 | 0.7 | 0 | 1.1 | 0 | 1.1 | 0 | 0.7 | 0 | 0.7 |
| 100 | 800 | 11.7 | 0.4 | 11.7 | 0.5 | 13.7 | 0.7 | 12.7 | 0.7 | 12 | 0.4 | 11.7 | 0.5 |
| 200 | 800 | 0 | 0.4 | 0 | 0.3 | 0 | 0.6 | 0 | 0.6 | 0 | 0.4 | 0 | 0.4 |
| 300 | 800 | 0 | 0.3 | 0 | 0.3 | 0 | 0.6 | 0 | 0.6 | 0 | 0.3 | 0 | 0.3 |
| | | $t_3$ Distribution | | | | | | | | | | | |
| 100 | 400 | 53.3 | 10.6 | 51.7 | 10.5 | 12 | 1.3 | 11.7 | 1.4 | 5 | 0.9 | 4 | 0.9 |
| 200 | 400 | 17 | 9.4 | 18.3 | 9.5 | 0 | 1.4 | 0 | 1.5 | 0 | 0.7 | 0 | 0.8 |
| 300 | 400 | 8.7 | 6.6 | 9 | 6.6 | 0 | 1.3 | 0 | 1.5 | 0 | 0.7 | 0 | 0.7 |
| 100 | 800 | 66.3 | 8.9 | 67.7 | 8.9 | 33.3 | 0.7 | 35.7 | 0.6 | 24.3 | 0.5 | 24.3 | 0.5 |
| 200 | 800 | 17 | 1.9 | 20.7 | 1.9 | 0 | 0.8 | 0 | 0.7 | 0 | 0.4 | 0 | 0.4 |
| 300 | 800 | 13.7 | 5.8 | 15 | 5.9 | 0 | 0.7 | 0 | 0.8 | 0 | 0.3 | 0 | 0.3 |
| | | Mixture Normal Distribution | | | | | | | | | | | |
| 100 | 400 | 55.7 | 2.1 | 59 | 1.5 | 28.3 | 1.3 | 25 | 1.5 | 7.3 | 0.9 | 7 | 1 |
| 200 | 400 | 10.3 | 1.9 | 8.7 | 1.7 | 1.3 | 1.8 | 1 | 1.7 | 0 | 0.8 | 0 | 0.7 |
| 300 | 400 | 2.3 | 1.7 | 1.3 | 1.8 | 0 | 1.8 | 0 | 1.7 | 0 | 0.7 | 0 | 0.6 |
| 100 | 800 | 75.7 | 1.5 | 80.3 | 1 | 58.7 | 0.9 | 58 | 0.8 | 36.3 | 0.5 | 37.3 | 0.5 |
| 200 | 800 | 22.7 | 1.1 | 20.7 | 1.1 | 4.7 | 1 | 5 | 0.9 | 3 | 0.3 | 3 | 0.4 |
| 300 | 800 | 2.3 | 0.9 | 1.7 | 0.8 | 0 | 0.9 | 0 | 0.9 | 0 | 0.3 | 0 | 0.3 |

oil diet "LIN" and fish oil diet "FISH"). We compare SSCCA to KSCCA and SCCA by focusing on the first canonical pair. In our analysis, we let $X_1$ (with dimension $p_1 = 120$) be the gene expression measurements, and $X_2$ (with dimension $p_2 = 21$) be the fatty acids. All three methods are implemented using the *findw12bic* function in the R package `mixedCCA`, with tuning parameters selected via the $\text{BIC}_1$ criterion due to its superior variable selection performance (Yoon et al., 2020).

Table 4: The average of absolute difference between expected out-of-sample correlation and true canonical correlation ($|\hat{\rho} - \rho|$) of each method (multiplied by 100) under Model II.

| | | Normal Distribution | | | $t_3$ Distribution | | | Mixture Normal Distribution | | |
|---|---|---|---|---|---|---|---|---|---|---|
| $n$ | $p$ | SCCA | KSCCA | SSCCA | SCCA | KSCCA | SSCCA | SCCA | KSCCA | SSCCA |
| 100 | 400 | 3.1 | 2.2 | 3.2 | 34.2 | 5.7 | 4.9 | 31.2 | 11.1 | 5.8 |
| 200 | 400 | 4 | 3.5 | 4 | 14.4 | 2.6 | 4 | 5.1 | 1.6 | 3.9 |
| 300 | 400 | 4.2 | 3.8 | 4.2 | 9.7 | 3.3 | 4.2 | 2.9 | 2.2 | 4.2 |
| 100 | 800 | 11.6 | 11 | 11.6 | 45.1 | 25.8 | 22 | 51.8 | 37.7 | 27.3 |
| 200 | 800 | 3.8 | 3.4 | 3.8 | 19.7 | 3.2 | 4.7 | 6.6 | 2.6 | 4.8 |
| 300 | 800 | 4 | 3.7 | 4 | 11.6 | 3 | 4 | 2.4 | 1.9 | 4 |

Table 5: The average prediction loss of each method (multiplied by 100) under Model II.

| | | SCCA | | KSCCA | | SSCCA | |
|---|---|---|---|---|---|---|---|
| $n$ | $p$ | $L(\boldsymbol{w}_1, \hat{\boldsymbol{w}}_1)$ | $L(\boldsymbol{w}_2, \hat{\boldsymbol{w}}_2)$ | $L(\boldsymbol{w}_1, \hat{\boldsymbol{w}}_1)$ | $L(\boldsymbol{w}_2, \hat{\boldsymbol{w}}_2)$ | $L(\boldsymbol{w}_1, \hat{\boldsymbol{w}}_1)$ | $L(\boldsymbol{w}_2, \hat{\boldsymbol{w}}_2)$ |
| | | Normal Distribution | | | | | |
| 100 | 400 | 0.7 | 0.8 | 1.5 | 1.5 | 0.7 | 0.7 |
| 200 | 400 | 0.2 | 0.2 | 0.6 | 0.5 | 0.2 | 0.2 |
| 300 | 400 | 0.1 | 0.2 | 0.4 | 0.3 | 0.1 | 0.2 |
| 100 | 800 | 10.7 | 10.5 | 11.6 | 11.3 | 10.7 | 10.4 |
| 200 | 800 | 0.3 | 0.3 | 0.6 | 0.5 | 0.3 | 0.3 |
| 300 | 800 | 0.2 | 0.2 | 0.4 | 0.4 | 0.2 | 0.1 |
| | | $t_3$ Distribution | | | | | |
| 100 | 400 | 30.2 | 32 | 5.8 | 6.2 | 2.7 | 2.7 |
| 200 | 400 | 13.8 | 13.7 | 1.2 | 1 | 0.2 | 0.2 |
| 300 | 400 | 7.8 | 7.9 | 0.7 | 0.7 | 0.1 | 0.1 |
| 100 | 800 | 44.7 | 43.5 | 28 | 27.4 | 22.6 | 22.4 |
| 200 | 800 | 19.4 | 18.9 | 2.2 | 2.2 | 1.3 | 1.3 |
| 300 | 800 | 10 | 10 | 0.8 | 0.8 | 0.2 | 0.2 |
| | | Mixture Normal Distribution | | | | | |
| 100 | 400 | 25.2 | 22.2 | 10.9 | 9.9 | 3.6 | 3.7 |
| 200 | 400 | 4 | 4.7 | 2.2 | 2.2 | 0.3 | 0.3 |
| 300 | 400 | 1.4 | 1.5 | 1.3 | 1.4 | 0.1 | 0.1 |
| 100 | 800 | 48.1 | 47.2 | 37.7 | 38.5 | 28.3 | 28.4 |
| 200 | 800 | 5.9 | 5.8 | 3.2 | 3.1 | 1.3 | 1.2 |
| 300 | 800 | 1.7 | 1.7 | 1.6 | 1.5 | 0.2 | 0.2 |

To evaluate the performance of different methods, we adopt a repeated random splitting strategy: the data are randomly split into two parts, one with 80% of the observations as the training set, and another with the remainder as the test set. We implement three methods on the training set to obtain $\hat{\boldsymbol{w}}_{1,\text{train}}$ and $\hat{\boldsymbol{w}}_{2,\text{train}}$ and compute the out-of-sample correlation

$$\hat{\rho}_{\text{test}} = \left| \frac{\hat{\boldsymbol{w}}_{1,\text{train}}^\top \boldsymbol{\Sigma}_{12,\text{test}} \hat{\boldsymbol{w}}_{2,\text{train}}}{(\hat{\boldsymbol{w}}_{1,\text{train}}^\top \boldsymbol{\Sigma}_{1,\text{test}} \hat{\boldsymbol{w}}_{1,\text{train}})^{1/2} (\hat{\boldsymbol{w}}_{2,\text{train}}^\top \boldsymbol{\Sigma}_{2,\text{test}} \hat{\boldsymbol{w}}_{2,\text{train}})^{1/2}} \right|$$

based on the test set, where $\Sigma_{\text{test}}$ denotes the covariance matrix estimated from the test data. For SSCCA, KSCCA, and SCCA, $\Sigma_{\text{test}}$ corresponds to the spatial-sign covariance matrix, Kendall's tau-based covariance estimator, and sample covariance matrix, respectively. This process is repeated

Table 6: The average of the false positive and false negative rates (multiplied by 100) of the selected model size of each method under Model II.

| | | SCCA | | | | KSCCA | | | | SSCCA | | | |
|---|---|---|---|---|---|---|---|---|---|---|---|---|---|
| $n$ | $p$ | $FPR_1$ | $FNR_1$ | $FPR_2$ | $FNR_2$ | $FPR_1$ | $FNR_1$ | $FPR_2$ | $FNR_2$ | $FPR_1$ | $FNR_1$ | $FPR_2$ | $FNR_2$ |
| | | | | | | Normal Distribution | | | | | | | |
| 100 | 400 | 0 | 0.8 | 0 | 0.9 | 0 | 1.3 | 0 | 1.3 | 0 | 0.8 | 0 | 0.9 |
| 200 | 400 | 0 | 0.9 | 0 | 0.9 | 0 | 1.5 | 0 | 1.5 | 0 | 0.9 | 0 | 0.9 |
| 300 | 400 | 0 | 0.9 | 0 | 0.8 | 0 | 1.5 | 0 | 1.4 | 0 | 0.9 | 0 | 0.9 |
| 100 | 800 | 10 | 0.5 | 9.7 | 0.5 | 10.7 | 0.7 | 10.7 | 0.6 | 10 | 0.5 | 9.7 | 0.5 |
| 200 | 800 | 0 | 0.4 | 0 | 0.4 | 0 | 0.8 | 0 | 0.8 | 0 | 0.4 | 0 | 0.4 |
| 300 | 800 | 0 | 0.4 | 0 | 0.5 | 0 | 0.8 | 0 | 0.8 | 0 | 0.4 | 0 | 0.4 |
| | | | | | | $t_3$ Distribution | | | | | | | |
| 100 | 400 | 43.7 | 12.2 | 45.7 | 11.5 | 6.7 | 1.5 | 6.3 | 1.4 | 2 | 1 | 2 | 0.8 |
| 200 | 400 | 13.3 | 9.4 | 13.7 | 9.4 | 0 | 1.7 | 0 | 1.6 | 0 | 0.9 | 0 | 0.8 |
| 300 | 400 | 7 | 6.3 | 6.7 | 7.3 | 0 | 1.7 | 0 | 1.6 | 0 | 0.8 | 0 | 0.8 |
| 100 | 800 | 54.7 | 9.8 | 55.7 | 9.7 | 32.3 | 0.7 | 28.3 | 0.7 | 22 | 0.5 | 21.7 | 0.5 |
| 200 | 800 | 20.7 | 5.7 | 21.3 | 5.8 | 1 | 0.9 | 1 | 0.9 | 1 | 0.4 | 1 | 0.4 |
| 300 | 800 | 10.3 | 5.7 | 10 | 5.7 | 0 | 1 | 0 | 0.9 | 0 | 0.4 | 0 | 0.5 |
| | | | | | | Mixture Normal Distribution | | | | | | | |
| 100 | 400 | 52.7 | 2 | 49.3 | 1.8 | 16.7 | 1.6 | 16.3 | 1.7 | 3 | 1 | 3 | 0.8 |
| 200 | 400 | 6.3 | 1.8 | 9.7 | 1.6 | 0 | 2 | 0.3 | 1.9 | 0 | 0.9 | 0 | 0.9 |
| 300 | 400 | 0.7 | 1.3 | 0.7 | 1.3 | 0 | 2 | 0 | 2.1 | 0 | 0.8 | 0 | 0.8 |
| 100 | 800 | 68.7 | 1.2 | 70.3 | 1 | 45 | 0.8 | 49.7 | 0.8 | 28 | 0.5 | 28 | 0.5 |
| 200 | 800 | 10 | 0.9 | 7.7 | 0.9 | 1.3 | 1 | 1 | 1 | 1 | 0.5 | 1 | 0.4 |
| 300 | 800 | 0.3 | 0.8 | 1 | 0.8 | 0 | 1.1 | 0 | 1.1 | 0 | 0.4 | 0 | 0.4 |

for 500 times. Table 7 summarizes the mean out-of-sample correlation and the average number of selected genes and fatty acids across different methods based on 500 replications.

From Table 7, we see that all three methods yield $\hat{\rho}_{\text{test}}$ values significantly different from zero, confirming their utility. However, SSCCA demonstrates superior performance, achieving the highest mean out-of-sample correlation while simultaneously selecting fewer variables on average compared to both KSCCA and SCCA. This combination of stronger predictive performance and sparser solutions underscores the advantages of SSCCA over KSCCA and SCCA in high-dimensional data analysis.

## 5 CONCLUSION

This paper proposes a robust sparse CCA method tailored for high-dimensional data under elliptical symmetric distributions. We establish the theoretical consistency of the estimator, demonstrating its reliability. Through extensive simulation studies and real data analysis, we show that the proposed method outperforms existing approaches, particularly in the presence of heavy-tailed distributions. Future research could extend SSCCA to handle structured sparsity or grouped variables, enhancing interpretability and performance with prior structural information. Adapting SSCCA to multi-set canonical correlation analysis (MCCA) would also be valuable for analyzing multimodal data. The primary limitation of our method lies in its reliance on the assumption of elliptical symmetric distributions. Exploring ways to relax this assumption and extend the methodology to accommodate more general or asymmetric distributions presents an important avenue for future research.

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

# Supplementary Material of "High Dimensional Sparse Canonical Correlation Analysis for Elliptical Symmetric Distributions"

The Supplementary Material contains additional simulation results and the proof of the theory part.

## A   ADDITIONAL SIMULATION RESULTS

Table 7 summaries the mean (standard deviation) of the out-of-sample correlation $\hat{\rho}_{\text{test}}$, the average number of selected genes, and the average number of selected fatty acids for SSCCA, KSCCA, and SCCA in the real-data example.

Table 7: The mean (standard deviation) of the out-of-sample correlation $\hat{\rho}_{\text{test}}$, the average number of selected genes, and the average number of selected fatty acids for SSCCA, KSCCA, and SCCA. In the parentheses are the standard deviations.

| Method | $\hat{\rho}_{\text{test}}$ | # selected genes | # selected fatty acids |
|--------|------------|------------------|------------------------|
| SSCCA | 0.762 (0.183) | 3.602 (1.442) | 2.812 (1.302) |
| KSCCA | 0.702 (0.203) | 3.992 (1.562) | 2.880 (1.317) |
| SCCA | 0.733 (0.211) | 3.936 (1.513) | 2.634 (1.204) |

We have conducted additional simulation studies comparing our proposed method, SSCCA, with several robust CCA variants to address the issue of non-normality and heavy-tailed distributions.

In the simulation study, in addition to KSCCA and SCCA which were compared in Section 3, we further add some other robust CCA variants, they are:

(1) the robust regression-based method proposed in Wilms & Croux (2016), which we denote as W&C method;

(2) the robust CCA methods proposed by Alfons et al. (2017b) which are widely used and can be easily implemented using the R package `ccaPP`. For ccaPP, we consider three alternatives which utilize Spearman's correlation, Kendall's tau, and Huber's M-estimator as CCA correlation measures, denoted as ccaPP-S, ccaPP-K, and ccaPP-M, respectively.

In Table 8, we present the results under the low-rank model (Model I in the simulation study) with $t_3$ distributed samples.

As indicated by the simulation results, SSCCA demonstrates the best performance among all the methods.

For the regression-based method W&C, we run the original code from the authors. Since the method W&C needs to solve the least-trimmed-square problem multiple times, it is very time-consuming and takes about tens of minutes to several hours to obtain the result for a single replication and several days for all the replications. Instead, our method returns the result within 5 seconds. Also, W&C does not perform well in the current settings, yielding a large average estimation error $|\hat{\rho} - \rho|$ and false positive and false negative rates.

Furthermore, ccaPP-S, ccaPP-K, and ccaPP-M also exhibit inferior performance. This outcome is expected, since these three approaches are not designed for variable selection and thus tend to perform poorly in high-dimensional contexts. Overall, the empirical results across multiple simulation settings consistently highlight the advantages of SSCCA in non-Gaussian and high-dimensional environments.

The following Table 9, studies the extremely challenging case when $\boldsymbol{X}$ is generated from the Cauchy distribution. From this table, we see that SSCCA demonstrates the best performance among all the methods, and SCCA breaks down in this setting.

We also conduct numerical experiments using the $L_0$ penalty. In the simulation study, we compare our proposed SSCCA method with SSCCA with $L_0$ penalty and KSCCA, which was compared in the original manuscript. In Table 10, we present the results under the low-rank model (Model I in the simulation study) with $t_3$ distributed samples.

Table 8: The average of absolute difference between expected out-of-sample correlation and true canonical correlation ($|\hat{\rho} - \rho|$) (multiplied by 100), the false positive rate (FPR) and false negative rate (FNR) for the estimated supports of the first (left) canonical vector, under Model I with $t_3$ distributed samples, based on 300 replications.

| Measure | $(n, p)$ | SSCCA | SCCA | KSCCA | W&C | ccaPP-S | ccaPP-K | ccaPP-M |
|---|---|---|---|---|---|---|---|---|
| $|\hat{\rho} - \rho|$ | (100, 400) | 6.7 | 44.7 | 23.4 | 61.1 | 62.8 | 61.8 | 72.7 |
| | (200, 400) | 0.8 | 18.3 | 2.4 | 20.4 | 31.8 | 29.3 | 35.8 |
| | (100, 800) | 23.7 | 54.6 | 40.3 | 81.1 | 78.4 | 78.2 | 89.9 |
| | (200, 800) | 0.7 | 18.2 | 2.5 | 46.1 | 41.0 | 40.0 | 43.9 |
| FPR(%) | (100, 400) | 5.0 | 53.3 | 12.0 | 46.7 | 13.3 | 24.3 | 11.0 |
| | (200, 400) | 0.0 | 17.0 | 0.0 | 18.0 | 1.0 | 1.3 | 0.3 |
| | (100, 800) | 24.3 | 66.3 | 33.3 | 71.3 | 37.7 | 51.3 | 33.3 |
| | (200, 800) | 0.0 | 17.0 | 0.0 | 72.0 | 10.67 | 14.00 | 9.3 |
| FNR(%) | (100, 400) | 0.9 | 10.6 | 1.3 | 9.1 | 51.0 | 37.2 | 76.9 |
| | (200, 400) | 0.7 | 9.4 | 1.4 | 3.2 | 66.8 | 53.3 | 98.0 |
| | (100, 800) | 0.5 | 8.9 | 0.7 | 6.9 | 31.3 | 23.5 | 46.6 |
| | (200, 800) | 0.4 | 1.9 | 0.8 | 3.9 | 52.6 | 40.8 | 79.3 |

Table 9: The average of absolute difference between expected out-of-sample correlation and true canonical correlation ($|\hat{\rho} - \rho|$) (multiplied by 100), the false positive rate (FPR) and false negative rate (FNR) for the estimated supports of the first (left) canonical vector, under Model I with Cauchy distributed samples, based on 300 replications.

| Measure | $(n, p)$ | SSCCA | SCCA | KSCCA | W&C |
|---|---|---|---|---|---|
| $|\hat{\rho} - \rho|$ | (100, 400) | 6.4 | – | 20.4 | 53.3 |
| | (200, 400) | 0.7 | – | 4.6 | 25.2 |
| | (100, 800) | 37.6 | – | 51.8 | 77.3 |
| | (200, 800) | 2.6 | – | 7.6 | 46.5 |
| FPR (%) | (100, 400) | 5.3 | – | 28.3 | 55.0 |
| | (200, 400) | 0.0 | – | 1.0 | 20.0 |
| | (100, 800) | 40.0 | – | 61.3 | 77.3 |
| | (200, 800) | 2.0 | – | 4.0 | 42.7 |
| FNR (%) | (100, 400) | 1.0 | – | 1.7 | 8.7 |
| | (200, 400) | 0.7 | – | 2.0 | 7.2 |
| | (100, 800) | 0.6 | – | 0.7 | 6.1 |
| | (200, 800) | 0.4 | – | 0.9 | 5.4 |

As indicated by the simulation results, the performance of SSCCA with $L_0$ penalty is inferior to SSCCA with $L_1$ penalty. This may be because the non-convex nature of the $L_0$ penalty, which makes the optimization procedure harder. Nevertheless, from a theoretical aspect, the $L_0$ penalty preserves several advantages, e.g., the stability and the pure sparse solutions. Therefore, the theory of the SSCCA method with the $L_0$ penalty and a more careful implementation of it, warrant further investigation.

In real data analysis, we also additionally compare SSCCA with W&C, ccaPP-S, ccaPP-K, and ccaPP-M. Table 11 summarizes the results. We can observe that all methods yield $\hat{\rho}_{\text{test}}$ values significantly different from zero, confirming their effectiveness. Among them, SSCCA exhibits superior performance, achieving the highest mean out-of-sample correlation while selecting fewer variables on average compared with other robust CCA variants. In contrast, W&C shows inferior predictive ability, with lower $\hat{\rho}_{\text{test}}$ and less sparse solutions than SSCCA. Notably, the three nonparametric

Table 10: The average of absolute difference between expected out-of-sample correlation and true canonical correlation ($|\hat{\rho} - \rho|$) (multiplied by 100), the false positive rate (FPR) and false negative rate (FNR) for the estimated supports of the first (left) canonical vector, under Model I with $t_3$ distributed samples, based on 300 replications.

| Measure | $(n, p)$ | SSCCA ($L_1$) | SSCCA ($L_0$) | KSCCA |
|---|---|---|---|---|
| $|\hat{\rho} - \rho|$ | $(100, 600)$ | 16.5 | 27.5 | 30.2 |
|  | $(200, 600)$ | 0.7 | 12.4 | 2.3 |
|  | $(300, 600)$ | 0.4 | 11.4 | 1.4 |
| FPR (%) | $(100, 600)$ | 17.0 | 30.3 | 25.3 |
|  | $(200, 600)$ | 0.0 | 27.0 | 0.0 |
|  | $(300, 600)$ | 0.0 | 29.7 | 0.0 |
| FNR (%) | $(100, 600)$ | 0.7 | 2.4 | 1.0 |
|  | $(200, 600)$ | 0.4 | 2.5 | 1.0 |
|  | $(300, 600)$ | 0.4 | 2.6 | 1.0 |

methods, ccaPP-S, ccaPP-K, and ccaPP-M, produce lower $\hat{\rho}_{\text{test}}$ values. These additional evidences further confirms SSCCA's superiority in practical scenarios involving potential non-normality.

Table 11: The mean (standard deviation) of the out-of-sample correlation $\hat{\rho}_{\text{test}}$, the average number of selected genes, and the average number of selected fatty acids for SSCCA, W&C, ccaPP-S, ccaPP-K, and ccaPP-M. In the parentheses are the standard deviations.

| Method | $\hat{\rho}_{\text{test}}$ | # selected genes | # selected fatty acids |
|---|---|---|---|
| SSCCA | 0.762 (0.183) | 3.602 (1.442) | 2.812 (1.302) |
| W&C | 0.681(0.222) | 9.260(2.200) | 5.224(1.592) |
| ccaPP-S | 0.572(0.245) | 41.584(7.675) | 8.408(2.451) |
| ccaPP-K | 0.574(0.229) | 27.550(6.245) | 6.400(2.258) |
| ccaPP-M | 0.483(0.207) | 56.034(26.069) | 3.788(3.290) |

## B    PROOF OF THEOREM 1

*Proof of Theorem 1.* By Lemma 1, we have $1 - \epsilon\|\hat{\boldsymbol{w}}_j\|_1^2 \leq \hat{\boldsymbol{w}}_j^\top \boldsymbol{\Sigma}_j \hat{\boldsymbol{w}}_j \leq 1 + \epsilon\|\hat{\boldsymbol{w}}_j\|_1^2$ for $j = 1, 2$, and

$$\hat{\boldsymbol{w}}_1^\top \boldsymbol{\Sigma}_{12} \hat{\boldsymbol{w}}_2 \geq \hat{\boldsymbol{w}}_1^\top (p\hat{\mathbf{S}}_{12})\hat{\boldsymbol{w}}_2 - |\hat{\boldsymbol{w}}_1^\top (p\hat{\mathbf{S}}_{12} - \boldsymbol{\Sigma}_{12})\hat{\boldsymbol{w}}_2| \geq \hat{\boldsymbol{w}}_1^\top (p\hat{\mathbf{S}}_{12})\hat{\boldsymbol{w}}_2 - \epsilon\|\hat{\boldsymbol{w}}_1\|_1\|\hat{\boldsymbol{w}}_2\|_1.$$

Let $\tilde{\boldsymbol{w}}_{1,*} = (\boldsymbol{w}_{1,*}^\top (p\hat{\mathbf{S}}_1)\boldsymbol{w}_{1,*})^{-1/2}\boldsymbol{w}_{1,*}$ and $\tilde{\boldsymbol{w}}_{2,*} = (\boldsymbol{w}_{2,*}^\top (p\hat{\mathbf{S}}_2)\boldsymbol{w}_{2,*})^{-1/2}\boldsymbol{w}_{2,*}$. By Assumptions 3 and Lemma 1, we have

$$1 - \epsilon\tau_1^2 \leq \boldsymbol{w}_{j,*}^\top (p\hat{\mathbf{S}}_j)\boldsymbol{w}_{j,*} \leq 1 + \epsilon\tau_1^2, j = 1, 2.$$

It follows that $\|\tilde{\boldsymbol{w}}_{j,*}\|_1 \leq (1 - \epsilon\tau_1^2)^{-1/2}\tau_1$, $j = 1, 2$. The following lower bound holds for the objective function at $(\tilde{\boldsymbol{w}}_{1,*}, \tilde{\boldsymbol{w}}_{2,*})$:

$$\begin{aligned}
&\tilde{\boldsymbol{w}}_{1,*}^\top (p\hat{\mathbf{S}}_{12})\tilde{\boldsymbol{w}}_{2,*} - \lambda\|\tilde{\boldsymbol{w}}_{1,*}\|_1 - \lambda\|\tilde{\boldsymbol{w}}_{2,*}\|_1 \\
\geq &\tilde{\boldsymbol{w}}_{1,*}^\top (p\hat{\mathbf{S}}_{12})\tilde{\boldsymbol{w}}_{2,*} - \lambda(1 - \epsilon\tau_1^2)^{-1/2}(\|\boldsymbol{w}_{1,*}\|_1 + \|\boldsymbol{w}_{2,*}\|_1) \\
\geq &\tilde{\boldsymbol{w}}_{1,*}^\top (p\hat{\mathbf{S}}_{12})\tilde{\boldsymbol{w}}_{2,*} - 2\lambda(1 - \epsilon\tau_1^2)^{-1/2}\tau_1 \\
\geq &\tilde{\boldsymbol{w}}_{1,*}^\top \boldsymbol{\Sigma}_{12}\tilde{\boldsymbol{w}}_{2,*} - |\tilde{\boldsymbol{w}}_{1,*}^\top (p\hat{\mathbf{S}}_{12} - \boldsymbol{\Sigma}_{12})\tilde{\boldsymbol{w}}_{2,*}| - 2\lambda(1 - \epsilon\tau_1^2)^{-1/2}\tau_1 \\
\geq &(1 + \epsilon\tau_1^2)^{-1}\rho_{1,*} - \epsilon(1 - \epsilon\tau_1^2)^{-1}\tau_1^2 - 2\lambda(1 - \epsilon\tau_1^2)^{-1/2}\tau_1.
\end{aligned}$$

Let $(\hat{\boldsymbol{w}}_1, \hat{\boldsymbol{w}}_2)$ be any pair such that

$$\hat{\boldsymbol{w}}_1^\top (p\hat{\mathbf{S}}_{12})\hat{\boldsymbol{w}}_2 - \lambda \|\hat{\boldsymbol{w}}_1\|_1 - \lambda \|\hat{\boldsymbol{w}}_2\|_1 \geq \tilde{\boldsymbol{w}}_{1,*}^\top (p\hat{\mathbf{S}}_{12})\tilde{\boldsymbol{w}}_{2,*} - \lambda \|\tilde{\boldsymbol{w}}_{1,*}\|_1 - \lambda \|\tilde{\boldsymbol{w}}_{2,*}\|_1. \tag{6}$$

we have

$$\hat{\boldsymbol{w}}_1^\top (p\hat{\mathbf{S}}_{12})\hat{\boldsymbol{w}}_2 - \lambda \|\hat{\boldsymbol{w}}_1\|_1 - \lambda \|\hat{\boldsymbol{w}}_2\|_1 \geq (1 + \epsilon\tau_1^2)^{-1}\rho_{1,*} - \epsilon(1 - \epsilon\tau_1^2)^{-1}\tau_1^2 - 2\lambda(1 - \epsilon\tau_1^2)^{-1/2}\tau_1.$$

Combining Lemma 1, we have

$$\hat{\boldsymbol{w}}_1^\top \boldsymbol{\Sigma}_{12}\hat{\boldsymbol{w}}_2 \geq (1 + \epsilon\tau_1^2)^{-1}\rho_{1,*} - \epsilon(1 - \epsilon\tau_1^2)^{-1}\tau_1^2 - 2\lambda(1 - \epsilon\tau_1^2)^{-1/2}\tau_1 - \epsilon\|\hat{\boldsymbol{w}}_1\|_1\|\hat{\boldsymbol{w}}_2\|_1, \tag{7}$$

and

$$\frac{\hat{\boldsymbol{w}}_1^\top \boldsymbol{\Sigma}_{12}\hat{\boldsymbol{w}}_2}{\sqrt{\hat{\boldsymbol{w}}_1^\top \boldsymbol{\Sigma}_1\hat{\boldsymbol{w}}_1}\sqrt{\hat{\boldsymbol{w}}_2^\top \boldsymbol{\Sigma}_2\hat{\boldsymbol{w}}_2}} \geq \frac{(1 + \epsilon\tau_1^2)^{-1}\rho_{1,*} - \epsilon(1 - \epsilon\tau_1^2)^{-1}\tau_1^2 - 2\lambda(1 - \epsilon\tau_1^2)^{-1/2}\tau_1 - \epsilon\|\hat{\boldsymbol{w}}_1\|_1\|\hat{\boldsymbol{w}}_2\|_1}{\sqrt{1 + \epsilon\|\hat{\boldsymbol{w}}_1\|_1^2}\sqrt{1 + \epsilon\|\hat{\boldsymbol{w}}_2\|_1^2}}. \tag{8}$$

We also have the upper bound result by Lemma 1:

$$\begin{aligned}\hat{\boldsymbol{w}}_1^\top (p\hat{\mathbf{S}}_{12})\hat{\boldsymbol{w}}_2 - \lambda \|\hat{\boldsymbol{w}}_1\|_1 - \lambda \|\hat{\boldsymbol{w}}_2\|_1 &\leq \hat{\boldsymbol{w}}_1^\top \boldsymbol{\Sigma}_{12}\hat{\boldsymbol{w}}_2 + \epsilon\|\hat{\boldsymbol{w}}_1\|_1\|\hat{\boldsymbol{w}}_2\|_1 - \lambda \|\hat{\boldsymbol{w}}_1\|_1 - \lambda \|\hat{\boldsymbol{w}}_2\|_1 \\ &\leq \rho_{1,*} + \epsilon\|\hat{\boldsymbol{w}}_1\|_1\|\hat{\boldsymbol{w}}_2\|_1 - \lambda \|\hat{\boldsymbol{w}}_1\|_1 - \lambda \|\hat{\boldsymbol{w}}_2\|_1.\end{aligned}$$

When (6) holds, the upper bound result implies that,

$$\rho_{1,*} + \epsilon\|\hat{\boldsymbol{w}}_1\|_1\|\hat{\boldsymbol{w}}_2\|_1 - \lambda \|\hat{\boldsymbol{w}}_1\|_1 - \lambda \|\hat{\boldsymbol{w}}_2\|_1 \geq (1 + \epsilon\tau_1^2)^{-1}\rho_{1,*} - \epsilon(1 - \epsilon\tau_1^2)^{-1}\tau_1^2 - 2\lambda(1 - \epsilon\tau_1^2)^{-1/2}\tau_1.$$

Rearranging the above inequality, we obtain

$$\lambda(\|\hat{\boldsymbol{w}}_1\|_1 + \|\hat{\boldsymbol{w}}_2\|_1) \leq B\tau_1^2\epsilon + \|\hat{\boldsymbol{w}}_1\|_1\|\hat{\boldsymbol{w}}_2\|_1\epsilon, \tag{9}$$

where

$$B = \frac{\rho_{1,*}}{1 + \tau_1^2\epsilon} + \frac{1}{1 - \tau_1^2\epsilon} + \frac{2C_1}{\sqrt{1 - \tau_1^2\epsilon}}.$$

Note that the above results hold generally for any pair $(\hat{\boldsymbol{w}}_1, \hat{\boldsymbol{w}}_2)$ that satisfies (6). We now specify the desired local maximizer for the theorem. Let $\lambda = C_1\tau_1\epsilon$ and $(\hat{\boldsymbol{w}}_1, \hat{\boldsymbol{w}}_2)$ be the global maximizer of the objective in (3) with the constraint $\max_{j\in\{1,2\}}\|\boldsymbol{w}_j\|_1 \leq C_2\tau_1$. As long as $C_2 \geq (1 - \tau_1^2\epsilon)^{-1/2}$, we have $\max_{j\in\{1,2\}}\|\boldsymbol{w}_{j,*}\|_1 \leq C_2\tau_1$ and the maximizer $(\hat{\boldsymbol{w}}_1, \hat{\boldsymbol{w}}_2)$ satisfies (6).

By choosing $C_1 \geq \frac{4}{\sqrt{1 - \tau_1^2\epsilon}} + \sqrt{\frac{20}{1 - \tau_1^2\epsilon} + \frac{4\rho_{1,*}}{1 + \tau_1^2\epsilon}}$, we have $C_1^2 \geq 4B$ and $[\max\{2BC_1^{-1}, (1 - \tau_1^2\epsilon)^{-1/2}\}, 2^{-1}C_1] \neq \emptyset$.

By choosing $C_2 \in [\max\{2BC_1^{-1}, (1 - \tau_1^2\epsilon)^{-1/2}\}, 2^{-1}C_1]$, we will show that $\max_{j\in\{1,2\}}\|\hat{\boldsymbol{w}}_j\|_1 < C_2\tau_1$. So $(\hat{\boldsymbol{w}}_1, \hat{\boldsymbol{w}}_2)$ will be a local maximizer of (3) without the constraint. Suppose, for the sake of contradiction, that $\|\hat{\boldsymbol{w}}_1\|_1 = C_2\tau_1$. It implies that $0 < \|\hat{\boldsymbol{w}}_2\|_1 \leq C_2\tau_1$. By (9), the following inequality must hold:

$$C_1C_2\tau_1^2\epsilon < (B + C_2^2)\tau_1^2\epsilon.$$

However, by the choice of $C_2$, we have $2^{-1}C_1C_2 \geq B$, $2^{-1}C_1C_2 \geq C_2^2$ and $C_1C_2 \geq B + C_2^2$. This contradicts the claim that (9) holds. Therefore with the choices of $C_1$ and $C_2$, we have $(\hat{\boldsymbol{w}}_1, \hat{\boldsymbol{w}}_2)$ is a local maximizer of (3) with $\max_{j\in\{1,2\}}\|\hat{\boldsymbol{w}}_j\|_1 \leq C_2\tau_1$ and (6) holds. Combing (7) and (8), we obtain the first two inequalities in the theorem,

$$\hat{\boldsymbol{w}}_1^\top \boldsymbol{\Sigma}_{12}\hat{\boldsymbol{w}}_2 \geq (1 + \tau_1^2\epsilon)^{-1}\rho_{1,*} - \{(1 - \epsilon\tau_1^2)^{-1} + 2C_1(1 - \epsilon\tau_1^2)^{-1/2} + C_2\}\tau_1^2\epsilon,$$

and

$$\frac{\hat{\boldsymbol{w}}_1^\top \boldsymbol{\Sigma}_{12}\hat{\boldsymbol{w}}_2}{\sqrt{\hat{\boldsymbol{w}}_1^\top \boldsymbol{\Sigma}_1\hat{\boldsymbol{w}}_1}\sqrt{\hat{\boldsymbol{w}}_2^\top \boldsymbol{\Sigma}_2\hat{\boldsymbol{w}}_2}} \geq \frac{(1 + \tau_1^2\epsilon)^{-1}\rho_{1,*} - \{(1 - \tau_1^2\epsilon)^{-1} + 2C_1(1 - \tau_1^2\epsilon)^{-1/2} + C_2\}\tau_1^2\epsilon}{1 + C_2\tau_1^2\epsilon}. \tag{10}$$

The last result follows from (10) and the following Lemma 2.

$\square$

**Lemma 2.** *Under Assumption 2, if the pair $(\boldsymbol{w}_1, \boldsymbol{w}_2)$ satisfies that*

$$\frac{\boldsymbol{w}_1^\top \boldsymbol{\Sigma}_{12} \boldsymbol{w}_2}{\sqrt{\boldsymbol{w}_1^\top \boldsymbol{\Sigma}_1 \boldsymbol{w}_1} \sqrt{\boldsymbol{w}_2^\top \boldsymbol{\Sigma}_2 \boldsymbol{w}_2}} \geq \rho_{1,*} - \delta,$$

*then*

$$\max_{j \in \{1,2\}} \cos^2(\angle(\boldsymbol{\Sigma}_j^{1/2} \boldsymbol{w}_j, \boldsymbol{\Sigma}_j^{1/2} \boldsymbol{w}_{j,*})) \leq 1 - \frac{2\delta}{\gamma}.$$

*Additionally if Assumption 1 holds, then*

$$\max_{j \in \{1,2\}} \cos^2(\angle(\boldsymbol{w}_j, \boldsymbol{w}_{j,*})) \leq 1 - \frac{4\kappa\delta}{\gamma}.$$

*Proof.* Lemma 2 is a direct consequence of Lemma 5 and Lemma 6 in Mai & Zhang (2019). We provide a neat and self-contained proof here for completeness.

Let $\{\boldsymbol{u}_k\}$ and $\{\boldsymbol{v}_k\}$ be the left and right singular vectors of $\boldsymbol{\Sigma}_1^{-1/2} \boldsymbol{\Sigma}_{12} \boldsymbol{\Sigma}_2^{-1/2}$. The singular value decomposition (SVD) is $\boldsymbol{\Sigma}_1^{-1/2} \boldsymbol{\Sigma}_{12} \boldsymbol{\Sigma}_2^{-1/2} = \sum_k \rho_{k,*} \boldsymbol{u}_k \boldsymbol{v}_k^\top$ and the canonical correlation vectors of $\boldsymbol{\Sigma}_{12}$ are $\{\boldsymbol{\Sigma}_1^{-1/2} \boldsymbol{u}_k\}$ and $\{\boldsymbol{\Sigma}_2^{-1/2} \boldsymbol{v}_k\}$. It means that $\boldsymbol{\Sigma}_{12} = \sum_k \rho_{k,*} \boldsymbol{\Sigma}_1^{1/2} \boldsymbol{u}_k \boldsymbol{v}_k^\top \boldsymbol{\Sigma}_2^{1/2}$, $\boldsymbol{w}_{1,*} = \boldsymbol{\Sigma}_1^{-1/2} \boldsymbol{u}_1$ and $\boldsymbol{w}_{2,*} = \boldsymbol{\Sigma}_2^{-1/2} \boldsymbol{v}_1$.

Denote the normalized vector $\bar{\boldsymbol{w}}_j = \|\boldsymbol{\Sigma}_j^{1/2} \boldsymbol{w}_j\|_2^{-1} \boldsymbol{\Sigma}_j^{1/2} \boldsymbol{w}_j$. By the decomposition of $\boldsymbol{\Sigma}_{12}$,

$$\frac{\boldsymbol{w}_1^\top \boldsymbol{\Sigma}_{12} \boldsymbol{w}_2}{\sqrt{\boldsymbol{w}_1^\top \boldsymbol{\Sigma}_1 \boldsymbol{w}_1} \sqrt{\boldsymbol{w}_2^\top \boldsymbol{\Sigma}_2 \boldsymbol{w}_2}} = \frac{\sum_k \rho_{k,*} \boldsymbol{w}_1^\top \boldsymbol{\Sigma}_1^{1/2} \boldsymbol{u}_k \boldsymbol{v}_k^\top \boldsymbol{\Sigma}_2^{1/2} \boldsymbol{w}_2}{\sqrt{\boldsymbol{w}_1^\top \boldsymbol{\Sigma}_1 \boldsymbol{w}_1} \sqrt{\boldsymbol{w}_2^\top \boldsymbol{\Sigma}_2 \boldsymbol{w}_2}}$$

$$\leq \sum_k \frac{\rho_{k,*}}{2} \{(\bar{\boldsymbol{w}}_1^\top \boldsymbol{u}_k)^2 + (\bar{\boldsymbol{w}}_2^\top \boldsymbol{v}_k)^2\}$$

$$\leq \frac{\rho_{1,*} \{(\bar{\boldsymbol{w}}_1^\top \boldsymbol{u}_1)^2 + (\bar{\boldsymbol{w}}_2^\top \boldsymbol{v}_1)^2\} + (\rho_{1,*} - \gamma)\{2 - (\bar{\boldsymbol{w}}_1^\top \boldsymbol{u}_1)^2 - (\bar{\boldsymbol{w}}_2^\top \boldsymbol{v}_1)^2\}}{2}$$

$$= \rho_{1,*} - \frac{\gamma\{2 - (\bar{\boldsymbol{w}}_1^\top \boldsymbol{u}_1)^2 - (\bar{\boldsymbol{w}}_2^\top \boldsymbol{v}_1)^2\}}{2}.$$

Therefore,

$$2 - (\bar{\boldsymbol{w}}_1^\top \boldsymbol{u}_1)^2 - (\bar{\boldsymbol{w}}_2^\top \boldsymbol{v}_1)^2 \leq \frac{2\delta}{\gamma}.$$

By the facts that $(\bar{\boldsymbol{w}}_1^\top \boldsymbol{u}_1)^2 = \cos^2(\angle(\bar{\boldsymbol{w}}_1, \boldsymbol{u}_1)) \in [0,1]$ and $(\bar{\boldsymbol{w}}_2^\top \boldsymbol{v}_1)^2 = \cos^2(\angle(\bar{\boldsymbol{w}}_2, \boldsymbol{v}_1)) \in [0,1]$, we have $\cos^2(\angle(\bar{\boldsymbol{w}}_1, \boldsymbol{u}_1)) \geq 1 - \frac{2\delta}{\gamma}$ and $\cos^2(\angle(\bar{\boldsymbol{w}}_2, \boldsymbol{v}_1)) \geq 1 - \frac{2\delta}{\gamma}$, i.e.,

$$\max_{j \in \{1,2\}} \cos^2(\angle(\boldsymbol{\Sigma}_j^{1/2} \boldsymbol{w}_j, \boldsymbol{\Sigma}_j^{1/2} \boldsymbol{w}_{j,*})) \geq 1 - \frac{2\delta}{\gamma}.$$

For the second result, let $\widetilde{\boldsymbol{w}}_j = \|\boldsymbol{\Sigma}_j^{1/2} \boldsymbol{w}_j\|_2^{-1} \boldsymbol{w}_j$. Note that

$$1 - \cos(\angle(\boldsymbol{w}_j, \boldsymbol{w}_{j,*})) = \frac{\|\widetilde{\boldsymbol{w}}_j\|_2 \|\boldsymbol{w}_{j,*}\|_2 - \widetilde{\boldsymbol{w}}_j^\top \boldsymbol{w}_{j,*}}{\|\widetilde{\boldsymbol{w}}_j\|_2 \|\boldsymbol{w}_{j,*}\|_2} \leq \frac{\|\widetilde{\boldsymbol{w}}_j\|_2^2 + \|\boldsymbol{w}_{j,*}\|_2^2 - 2\widetilde{\boldsymbol{w}}_j^\top \boldsymbol{w}_{j,*}}{2\|\widetilde{\boldsymbol{w}}_j\|_2 \|\boldsymbol{w}_{j,*}\|_2}$$

$$= \frac{(\widetilde{\boldsymbol{w}}_j - \boldsymbol{w}_{j,*})^\top (\widetilde{\boldsymbol{w}}_j - \boldsymbol{w}_{j,*})}{2\|\widetilde{\boldsymbol{w}}_j\|_2 \|\boldsymbol{w}_{j,*}\|_2} \leq \frac{\kappa(\widetilde{\boldsymbol{w}}_j - \boldsymbol{w}_{j,*})^\top \boldsymbol{\Sigma}_j (\widetilde{\boldsymbol{w}}_j - \boldsymbol{w}_{j,*})}{2\|\boldsymbol{\Sigma}_j^{1/2} \widetilde{\boldsymbol{w}}_j\|_2 \|\boldsymbol{\Sigma}_j^{1/2} \boldsymbol{w}_{j,*}\|_2}$$

$$= \kappa(1 - \widetilde{\boldsymbol{w}}_j^\top \boldsymbol{\Sigma}_j \boldsymbol{w}_{j,*}) = \kappa\{1 - \cos(\angle(\boldsymbol{\Sigma}_j^{1/2} \widetilde{\boldsymbol{w}}_j, \boldsymbol{\Sigma}_j^{1/2} \boldsymbol{w}_{j,*}))\}$$

$$= \kappa\{1 - \cos(\angle(\boldsymbol{\Sigma}_j^{1/2} \boldsymbol{w}_j, \boldsymbol{\Sigma}_j^{1/2} \boldsymbol{w}_{j,*}))\},$$

where in the second last inequality we use Assumption 1 and the property that $\|\widetilde{\boldsymbol{w}}_j\|_1 \leq C_2 \tau_1$ for some $C_2 > 0$ from the proof of Theorem 1, and in the second last identity we use the fact that $\|\boldsymbol{\Sigma}_j^{1/2} \boldsymbol{w}_{j,*}\|_2 = \|\boldsymbol{\Sigma}_j^{1/2} \widetilde{\boldsymbol{w}}_j\|_2 = 1$. Therefore,

$$\cos(\angle(\boldsymbol{w}_j, \boldsymbol{w}_{j,*})) \geq 1 - \kappa(1 - \sqrt{1 - 2\delta\gamma^{-1}}),$$

and finally,

$$\cos^2(\angle(\boldsymbol{w}_j, \boldsymbol{w}_{j,*})) \geq 1 - 2\kappa(1 - \sqrt{1 - 2\delta\gamma^{-1}}) = 1 - \frac{4\kappa\delta\gamma^{-1}}{1 + \sqrt{1 - 2\delta\gamma^{-1}}} \geq 1 - \frac{4\kappa\delta}{\gamma}.$$

$\square$

