# OpenReview forum: "High Dimensional Sparse Canonical Correlation Analysis for Elliptical Symmetric Distributions"
_ICLR.cc/2026/Conference — ICLR 2026 Conference Withdrawn Submission_

### Official Review · Reviewer_rczW · 2025-10-16

**Soundness:** 3
**Presentation:** 2
**Contribution:** 2
**Rating:** 4
**Confidence:** 3

**Summary:**

The paper proposes a variant of sparse canonical correlation analysis (CCA) in which the covariance matrix is estimated by the spatial-sign covariance. The spatial-sign covariance is a simple robust estimator of the covariance in which only the scaled direction from the median is used. The authors analyze theoretical property of the proposed sparse CCA. Under the non-Gaussian assumption, the convergence rate is theoretically established. The empirical evaluation is performed for both on synthetic and real datasets.

**Strengths:**

Overall, technical quality is seemingly fine. Although I couldn't fully follow the entire proof, the theoretical result seems reasonable.

The experimental results appear to be consistent with the basic claims.

Although neural based approaches have become a mainstream and other classical approaches tend to be overlooked, I think researches on interpretable approaches like this paper remains important.

**Weaknesses:**

The technical novelty of outside the theoretical analysis is somewhat weak (and seemingly, only theorem 1 is the original theoretical analysis, though result itself is reasonable). Replacing covariance of sparse CCA with spatial-sign covariance is interesting, but the formulation itself is straightforward (seemingly, a direct extension of the counterpart of sparse PCA) and the optimization algorithm is also an existing method.

Theoretical justification of BIC is not explicitly revealed (In theoretical analysis, lambda is not determined by BIC).

Some technical descriptions are a bit confusing.

**Questions:**

Does the condition 'tau_1^2 (\sqrt{(log p)/n} + p^{-1/2}) -> 0' mean the convergence cannot be guaranteed for the fixed p setting?

Is there anything that can be clarified about support recovery based on the current theoretical analysis?

BIC is separately defined for lambda_1 and lambda_2, but then, how the 'combination' of them can be selected? (adding BIC for lambda1 and lambda2?)

I'm a bit confused about 'p' in (3). What is definition of p? After (2), it is defined as E(r^2) = p and also used in the sphere S^p-1 of u. On the other hand, p seems a dimension of concatenated vector in the definition of \hat{\mu} (Further, notation X_i is confusing, because X_1 and X_2 are first defined as vectors of two information sources of CCA while X_i also indicates the i-th observation of the concatenated vector in the later discussion). Then, my current guess is that p = p_1 + p_2 and the same value is also used in the dimension of the (latent) sphere and the expectation of r? If so, what is the rational behind this setting?

---

### Official Review · Reviewer_fbhp · 2025-10-29

**Soundness:** 3
**Presentation:** 3
**Contribution:** 2
**Rating:** 2
**Confidence:** 4

**Summary:**

This paper deals with the problem of extracting canonical correlations from a set of pairs of high-dimensional examples, and proposes a method that introduces the spatial-sign covariance matrix instead of standard covariance matrices. Since the spatial-sign covariance matrix checks only a direction (more specifically a sign of each dimension) from a mean or a median to every example, the proposed method is robust against outliers in terms of norms. The proposed method also incorporates L1 sparsity of bases and BIC-based hyper-parameter tuning. This paper demonstrates that under several assumptions for bases and singular values the proposed method asymptotically produces the same outcome as the standard canonical correlations analysis (with standard covariance matrices).

**Strengths:**

S1. The problem dealt with in this paper is significant in pattern recognition and computer vision. Extracting canonical correlations is a fundamental task in statistics, and it has become established as a major tool in pattern recognition. However, as presented in this paper, it faces difficulty for high-dimensional low-resource datasets since CCA builds on linear matrix decomposition. This paper tries to overcome this problem with spatial-sign covariance matrices and sparsity-induced penalty, which is a reasonable choice in terms of simplicity.

S2. This paper provides theoretical analysis related to convergence of the proposed method to the standard CCA under several assumptions, which would be preferable for researchers and engineers familiar with the standard CCA.

S3. The current paper is basically well-written and easy to follow.

**Weaknesses:**

W1. I could not understand the novelty of the proposed method against several previous methods.
  - As presented in the introduction, several methods for PCA with spatial sign correlation matrices have already been proposed, and unfortunately some of them have already been published, such as https://www.sciencedirect.com/science/article/abs/pii/S0167715212000028 and https://onlinelibrary.wiley.com/doi/abs/10.1111/biom.13695 . PCA and CCA are closely related to each other and share the same methodology in terms of matrix decomposition, which means that just applying the idea presented in the context of PCA to CCA might be new but not sufficiently novel.
  - Also, this paper does not bring any other novel components specific for CCA. Introducing L1-sparsity and BIC-based parameter tuning is frequently applied to various multivariate analysis including PCA and CCA.
  - If the authors believe that the proposed method has sufficiently novel components, the authors should have justified its novelty and demonstrated its effectiveness by empirical comparisons with the method excluding it.

**Questions:**

Q1. [Minor] Line 249: Y seems to be undefined. Is this a typo of “X1 and X2”?

---

### Official Review · Reviewer_htyz · 2025-10-30

**Soundness:** 2
**Presentation:** 2
**Contribution:** 2
**Rating:** 2
**Confidence:** 4

**Summary:**

This paper addresses critical limitations of classical Canonical Correlation Analysis (CCA) in high-dimensional, non-normal data by proposing a robust sparse method (SSCCA) tailored for elliptical symmetric distributions. Classical CCA fails here due to ill-conditioned sample covariance matrices (high dimensions) and sensitivity to heavy tails/outliers (non-normality); existing sparse CCA methods (e.g., SCCA, KSCCA) still rely on fragile second-order moment assumptions.

SSCCA’s core innovation is replacing sample covariance matrices with spatial-sign covariance matrices (robust to heavy tails, no finite higher-moment requirement) and adding \(\ell_1\) penalties to induce sparsity. Its optimization problem maximizes \(w_1^\top p\hat{S}_{12}w_2 - \lambda_1\|w_1\|_1 - \lambda_2\|w_2\|_1\) (with \(p\hat{S}_{1}, p\hat{S}_{2}\) as scaled spatial-sign covariance estimates) and uses BIC-like criteria for tuning.

Key Contributions.

1. SSCCA handles high dimensions and elliptical heavy-tailed data (e.g., t-distribution, mixed normals) where existing methods fail, with no finite moment assumption.

2. Under mild conditions (bounded eigenvalues, sparse canonical vectors), SSCCA is consistent.

3. Simulations (3 distributions, 3D/4D settings) show SSCCA outperforms SCCA/KSCCA.

4. A real nutrimouse data application (120 genes, 21 fatty acids) confirms SSCCA achieves the highest out-of-sample correlation (0.762) with sparser variable selection (3.6 genes, 2.8 fatty acids on average).

**Strengths:**

The paper demonstrates notable originality by addressing a critical dual gap in existing CCA literature: simultaneously robustness to heavy-tailed non-normal data and adaptability to high-dimensional settings—a challenge that has been unmet by classical or most modern sparse CCA methods. Unlike prior works (e.g., SCCA relying on sample covariance, KSCCA using Kendall’s tau), it innovatively integrates spatial-sign covariance matrices (resistant to heavy tails, no finite higher-moment requirement) with \(\ell_1\) regularization for sparsity, specifically tailoring the framework to elliptical symmetric distributions. Therefore, the setting considered in the paper is new.

**Weaknesses:**

My biggest concern is its technical novelty. Although the setting is new, the paper seems to directly use the results from paper,
Zhengke Lu and Long Feng. Robust sparse precision matrix estimation and its application. arXiv preprint arXiv:2503.03575, 2025, to prove its main theorem.

**Questions:**

1. Could authors provide more motivations why we should consider SPARSE CANONICAL CORRELATION ANALYSIS problem? It seems like the technique of using sparse-sign covariance matrix can be used in many other types of problems.

2. What is the motivation to consider the setting where data are generated from the elliptical symmetric distribution? What is the difficulty of extending it to other distributions?

3. According to the equation (2), it seems to me that the covariance of X is p \Gamma \Gamma^T not \Gamma \Gamma^T?

4. In Lemma 1, the bound of $\|p \hat S - \Sigma \|_{\infty}$ has two terms, $O(\sqrt{\log p / n})$ and $O(1/\sqrt{p})$. What is the intuition of the second term? Why does the error decrease with the order $\sqrt{p}$, as $p \rightarrow \infty$?

5. Could the author summarize the technical novelty and difficulty in the manuscript?

**Details Of Ethics Concerns:**

No Concern.

---

### Official Review · Reviewer_yhpz · 2025-11-02

**Soundness:** 3
**Presentation:** 3
**Contribution:** 2
**Rating:** 4
**Confidence:** 4

**Summary:**

This paper proposes a robust high-dimensional sparse canonical correlation analysis method for investigating linear relationships between two high-dimensional random vectors, focusing on elliptical symmetric distributions. The authors replace sample covariance matrices with spatial-sign covariance matrices, which are more robust to heavy-tailed distributions, and establish theoretical consistency results.

**Strengths:**

1. The paper addresses a genuine limitation of classical CCA—its poor performance under heavy-tailed distributions and high dimensionality. The use of spatial-sign covariance matrices is well-motivated for elliptical symmetric distributions and yields a robust formulation without needing finite higher moments.

2. The simulation studies are extensive, comparing against multiple baselines across different distributions.

**Weaknesses:**

1. Theorem 1 establishes properties for the existence of a suitable local maximizer of Problem (3), but the paper uses mixedCCA’s routine without guarantees it converges to such a point. No convergence guarantees are shown for the implemented solver.

2. The derived rate in Theorem 1 includes an additional p^(−1/2) term compared to prior work. While the authors argue this term becomes negligible, they provide no justification for whether this rate is minimax optimal or if the extra term is an artifact of the proof technique.

**Questions:**

See the weakness.

---

### Official Review · Reviewer_4jQz · 2025-11-05

**Soundness:** 3
**Presentation:** 3
**Contribution:** 3
**Rating:** 4
**Confidence:** 4

**Summary:**

This paper proposes a robust high-dimensional sparse canonical correlation analysis (SSCCA) method tailored for elliptical symmetric distributions, addressing limitations of traditional CCA methods that struggle with high-dimensionality and heavy-tailed data. The key innovation is the use of the spatial-sign covariance matrix as a robust estimator, combined with an ℓ₁ penalty to induce sparsity. Theoretical guarantees for consistency and robustness are established, and simulations demonstrate that SSCCA outperforms existing methods—especially under heavy-tailed distributions—in estimation accuracy, prediction loss, and variable selection. A real-data application further validates its practical utility, offering a stable and efficient alternative for high-dimensional data analysis.

**Strengths:**

1. It introduces a novel and effective solution for high-dimensional data by creatively combining the robustness of spatial-sign covariance matrices with sparse CCA, specifically targeting the common challenge of heavy-tailed distributions.
2. The proposed method is rigorously validated through a comprehensive theoretical analysis proving its consistency and extensive simulations demonstrating its superior performance against state-of-the-art alternatives.
3. The work has significant practical impact, providing a powerful and computationally efficient tool for integrative analysis in fields like genomics and finance where high-dimensional, non-Gaussian data is the norm.

**Weaknesses:**

1. The method's theoretical guarantees and optimal performance are explicitly contingent on the assumption of elliptical symmetry, which may be restrictive and is often difficult to verify in real-world applications.
2. The paper does not deeply explore the computational complexity or scalability of the proposed algorithm, especially for ultra-high-dimensional problems (e.g., p>>10,000), beyond noting its use of existing convex optimization routines.
3. The real-data analysis, while demonstrating utility, is limited to a single, relatively small-scale biological dataset, leaving its performance on larger, noisier, or fundamentally different data types (e.g., text or network data) an open question.
4. the work is too old, how it can connect with the state-of-the-art deep learning based methods.

**Questions:**

1. The method's theoretical guarantees and optimal performance are explicitly contingent on the assumption of elliptical symmetry, which may be restrictive and is often difficult to verify in real-world applications.
2. The paper does not deeply explore the computational complexity or scalability of the proposed algorithm, especially for ultra-high-dimensional problems (e.g., p>>10,000), beyond noting its use of existing convex optimization routines.
3. The real-data analysis, while demonstrating utility, is limited to a single, relatively small-scale biological dataset, leaving its performance on larger, noisier, or fundamentally different data types (e.g., text or network data) an open question.
4. the work is too old, how it can connect with the state-of-the-art deep learning based methods.

---

### Note · Authors · 2025-11-17

I have read and agree with the venue's withdrawal policy on behalf of myself and my co-authors.